# Recent Advances in the Chemistry of Saturated Annulated Nitrogen-Containing Polycyclic Compounds

**DOI:** 10.3390/ijms232415484

**Published:** 2022-12-07

**Authors:** Victor Yu. Kirsanov, Elena B. Rakhimova

**Affiliations:** Institute of Petrochemistry and Catalysis, Ufa Federal Research Center of Russian Academy of Sciences, Prospekt Oktyabrya 141, 450075 Ufa, Russia

**Keywords:** nitrogen heterocycles, cycloreversion, biological activity

## Abstract

This review is devoted to the analysis of works published over the past 20 years on the chemistry of saturated annulated nitrogen-containing polycyclic compounds, the molecules of which consist of four, five, six, and seven cycles, and contain from one to eight endocyclic nitrogen atoms.

## 1. Introduction

Nitrogen compounds hold a special position among natural and synthetic organic substances. Nitrogen is present in important natural molecules: amino acids, proteins, nucleic acids, hormones, alkaloids, etc. Nitrogen compounds are significant for technology, agriculture, and medicine. Nitrogen is present in the molecules of numerous medicinal agents, organic fertilizers, polymer materials, and other valuable products. The number of new molecules containing several nitrogenous heterocyclic rings and possessing useful pharmacological properties is continuously increasing. In the next decade, the proportion of new nitrogen-containing pharmaceutical agents is expected to increase [1].

The current periodical literature contains numerous scientific publications devoted to the synthesis of unsaturated polyazapolycycles with different positions of nitrogen atoms at the periphery of the polycyclic skeleton [2]. However, there are markedly fewer publications dealing with saturated polyheterocyclic systems. In this review, we discuss and systematize the data published over the last 20 years on saturated *N*-containing polycyclic systems; the interest in these compounds is caused by their broad applicability in medical practice. We consider approaches to the targeted synthesis of tetra-, penta-, and hexa-cyclic saturated compounds containing one to several endocyclic nitrogen atoms in the molecule. As a rule, these molecules are designed using reactions such as cycloaddition, multicomponent cyclocondensation, and intramolecular cyclization, and the reduction of aza-aromatic systems to their saturated analogues. The review pays attention to some representatives of alkaloids, that is, *N*-containing polycyclic compounds of natural origin. Data on the biological activity found in some saturated *N*-containing polycyclic compounds are given.

## 2. Tetracyclic Saturated Nitrogen-Containing Compounds

### 2.1. Tetracyclanes with One Endocyclic Nitrogen Atom

Among saturated tetracyclic compounds with one endocyclic nitrogen atom, a mention should be made of natural compounds corresponding to the class of alkaloids. An example is (−)-lycopodine, the molecule of which is a 6/6/6/6 annulated polycyclic system. R.G. Carter and co-workers were the first to carry out the total enantioselective synthesis of (−)-lycopodine [3,4]. The intramolecular Michael reaction (C-7 and C-8) of keto sulfone **1,** and the subsequent Staudinger reduction of chiral azide **2** gave precursor **3**. The organocatalyzed Mannich cyclization (C-4 and C-13) of precursor **3** afforded azatricyclane **4** in 54% yield. The ring formation was accompanied by 1,3-rearrangement of sulfonic group from position C-8 to C-14. The subsequent sequence of steps starting from **4** resulted in the formation of target (−)-lycopodine **5** (Figure 1).

Like (−)-lycopodine **5**, (+)-clavolonine **10** is also a *Lycopodium* alkaloid. The difference between azatetracyclanes **5** and **10** is in that the latter has a hydroxyl group in position C-6. Despite the structural similarity of these two derivatives, their total syntheses proceed via different intermediates, but include similar chemical reactions [5], particularly, the Staudinger reduction of azide **6** to give imine **7** and intramolecular Mannich cyclization of imine **7** to tricyclane **8**. The Mannich reaction is preceded by epimerization of imine **7** on treatment with HCl in methanol. The subsequent nucleophilic substitution results in replacement of the –OMe group in tricyclane **8** by halogen, while alkaline hydrolysis of product **9** enables the synthesis of (+)-clavolonine **10** in 10% yield (Figure 2).

A number of representatives of *Lycopodium* alkaloids such as fawcettimine, lycoflexine, and serratinine type alkaloids also deserve attention, due to their unique tetracyclic structures containing two contiguous quaternary stereocenters and interesting biogenetic interconnections within this intriguing family. The research group headed by Yu-Rong Yang proposed [6] a general strategy for the total synthesis of (+)-fawcettimine **18**, (+)-lycoflexine **20**, and serratinine **24**. The key steps of this strategy include the highly efficient Helquist annulation to assemble the *cis*-fused carbocycle **13**; facile construction of aza-C_9_-membered derivatives **15** and **16** by double *N*-alkylation strategy; and SmI_2_ reduction of hydroxylated tricyclane **17** followed by *Boc*-deprotection and epimerization to (+)-fawcettimine **18**. The (+)-fawcettimine **18** synthesized in this way was transformed into (+)-lycoflexine **20** in 95% yield by intramolecular Mannich cyclization. For successful synthesis of serratinine type alkaloid **24**, the authors used asymmetric Shi epoxidation followed by post-treatment of *β*-epoxide **23** (Figure 3). Other approaches to the synthesis of related structures were also reported [7,8,9,10].

### 2.2. Tetracyclanes with Two Endocyclic Nitrogen Atoms

A facile two-step synthesis of saturated tetracycline **28** with two endocyclic nitrogen atoms has been proposed [11]. The reaction of D-(–)-camphorquinone **25** with *trans*-(*R,R*)-1,2-diaminocyclohexane **26** on refluxing in toluene (*p*-toluenesulfonic acid, 10% mol) furnished decahydrophenazine **27**. NaBH_4_ reduction of this product afforded chiral tetradecahydrophenazine **28** in 85% yield (Figure 4).

Enantiopure amino acids **29** and **30 a–c** were used as the initial building blocks for the synthesis of unsymmetrical tetracyclic diketopiperazines **31 a–c** [12]. The solubility of amino acids was increased by adding an ionic liquid to the reaction medium. On refluxing in toluene for 5 h, target heterodimers were obtained in 17–53% yields (Figure 5). 

*Matrine* type alkaloids **32**, the polycyclic skeleton of which is formed by two fused quinolizidine moieties A/B and C/D and contains four stereocenters at C-5, C-6, C-7, and C-11, were isolated from the *Oxytropisochrocephala* plant [13,14,15]. Cheng-jian Tan and co-workers studied the antiproliferative activity of these *bis*-quinolizidines against five human tumor cell lines (A549, KB, KB-VIN, MDA-MB-231, and MCF7) [16]. These compounds were also shown to suppress the expression of HSC70 protein, which plays a key role in the replication of hepatitis B virus [17]. In 2022, Chinese scientists were able to isolate ochrocephalamines E **33** [18]. This is the first example of quinolizidine alkaloid with a 6/6/6/5 polycyclic system (Figure 1).

Owing to the efforts of an international research group from France and Vietnam [19], total synthesis of tetracyclic (–)-schoberine **39,** as a representative of *Myrioneuron* alkaloids, was accomplished. Their structural peculiarity is the presence of the decahydroquinoline (DHQ) core. The subsequent attachment of rings C and D to the DHQ core (rings A and B) consistently generated the schoberine skeleton with a series of chiral centers. Cyclic alcohol **34** was converted to (–)-myrionine **37** via a sequence of reactions. The intramolecular cyclization of **37** induced by POCl_3_ in refluxing toluene followed by treatment with NaOH in methanol gave iminium **38**. The reduction of **38** by LiAlH_4_ in THF afforded **39** in a good yield (87%) (Figure 6).

Examples of the synthesis of other *Myrioneuron* alkaloids (Figure 2) have been described [20,21]. Schoberine B was evaluated in vitro against hepatitis C virus (HCV).

Przyby and co-worker [22] reported a method for isolation of (±)−lupanine, an alkaloid with a tetracyclic *bis*-quinolizine cage, and subsequent separation of the racemate into optically pure enantiomers. The LiAlH_4_ reduction of (−)−lupanine **40** enabled the synthesis of (+)−sparteine **41** (Figure 7), which is of interest for its pharmacological activity and as a chiral ligand for asymmetric synthesis.

### 2.3. Tetracyclanes with Four Endocyclic Nitrogen Atoms

The authors [23,24] demonstrated the possibility of preparing annulated tetraazatricyclanes, *cis/trans*-octahydrotetraazanaphthalenes **43, 44,** and *cis/trans*-decahydrodiimidazopyrazines **45, 46**, by condensation of triethylenetetramine **42** with glyoxal in water at 5 °C in the presence of Ca(OH)_2_ (Figure 8). The ratio and the yield of the resulting tricyclanesare affected by both replacement of the solvent and increase in the reaction temperature [25].

In continuation of this study, Italian research group headed by M. Argese proposed [26] a synthetic route to octahydrotetraazacyclopenta[*f,g*]acenaphthalene-1,2(3,4)-diones. A mixture of triazatricyclanes **43-46** was converted to isomeric diketones **47-50** by treatment with diethyl oxalate in EtOH at room temperature. The subsequent reduction by sodium dihydro-*bis*-(2-methoxyethoxy)aluminate (SBAH) [27] enabled the synthesis of previously unknown *cis/trans*-isomers of decahydrotetraazacyclopenta[*f,g*]acenaphthalenes **51** and **52** (Figure 8).

Unlike the Italian researchers cited above, P. Desogere and co-worker [28] used *N,N′*-*bis*-(aminoethyl)propane-1,3-diamine **53** as a linear tetramine. The condensation of tetramine **53** with glyoxal in ethanol at 0 °C resulted in the formation of tetraazatricyclane **54**. The reaction of the obtained *bis*-aminal with chloroacetaldehyde in the presence of sodium cyanide afforded isomeric polycyclic compounds **55** and **56** in a total yield of 42% (Figure 9).

The condensation of 1,4,5,8-tetraazadecalin **57** [29] with methyl acrylate in methanol gave stereoisomeric tetraazaperhydropyrenediones **58**-61 (Figure 10). The four isomers were formed in 20:16:35:29 ratio, with their total yield being 100%.

The same research group synthesized [30] non-trivial cage *bis*-aminals **63** using cyclohexane-1,2-dione **62** as the electrophilic agent. Cyclocondensation of *bis*-aminals **63** with dihaloalkanes **64** resulted in the formation of polycyclanes **65 a–c** in 80–90% yields (Figure 11).

The reactions of perhydro-2,2′-biperimidine **66** with glyoxal and ethyl glyoxalate have been studied [31]. In the former case, the reaction gave diastereomeric 4,5,9,10-tetramethoxytetraazaperhydropyrenes **68 a,b** in a total yield of 35%, while in the latter case, 5,10-dimethoxytetraazaperhydropyrenedione **67** was obtained (Figure 12).

A Japanese research group headed by T. Okawara proposed [32] a method for the synthesis of tetraazaperhydropyrenes, which was based on the reaction of tetramine **69** with two equivalents of glyoxal and benzotriazole. The reduction of dibenzotriazoyltetraazaperhydropyrene **70** afforded a mixture of *cis-* and *trans-*tetraazaperhydropyrenes **71 a,b** in 3% and 50% yields, respectively (Figure 13).

It was shown [33] that new enantiomerically pure 1,8,10,12-tetraazatetracyclopentadecanes **72** and **73** can be obtained by three-component condensation of (*R,R*)- and (*S,S*)-*trans*-1,2-diaminocyclohexane with paraformaldehyde and ammonia (Figure 14).

### 2.4. Tetracyclanes with Five Endocyclic Nitrogen Atoms

Tetracyclane **75** can be readily prepared in 85% yield by cyclization of 1,4,7,10,13-pentaazacyclopentadecane **74** with glyoxal in methanol at 20 °C (Figure 15). The subsequent transformation of **75** enabled the synthesis of ethylene cross-bridged tetraazamacrocycle used as an efficient ligand for transition metals [34].

### 2.5. Tetracyclanes with Six Endocyclic Nitrogen Atoms

The synthesis of tetracyclic glycoluril derivatives **77** and **78 a,b** [35,36], in particular, macrocyclic polyamines [37] has been reported. These tetracyclic compounds **77** and **78 a,b** were prepared by three-component cyclocondensation of tetrahydroimidazo [4,5-*d*]imidazole-2,5-diones (glycolurils) **76 a,b** with formaldehyde and structurally diverse primary amines (Figure 16).

An alternative method for the synthesis of hexaazatetracyclanes **79** [38] is based on the catalytic heterocyclization of glycoluril **76 a** with *N,N-bis*-(methoxymethyl)-*N*-alkylamines (Figure 17).

Tetrakis(hydroxymethyl)-substituted glycoluril **80** was used as an efficient synthon [39,40] for the synthesis of 2,6-disubstituted hexaazahexahydrocyclopentafluorene-4,8-diones **81** (Figure 18).

The principle of three-component cyclocondensation of monothioglycoluril **82** with paraformaldehyde and structurally diverse primary amines was implemented [41] for the synthesis of tetraazatetracyclanes **83** (Figure 19). Primary screening of the obtained compounds for antimicrobial activity revealed activity of the product with an ethyl substituent against the fungus *Cryptococcus neoformans*.

The authors have developed [42,43] a novel and efficient method for the selective synthesis of 2,7-di(cyclo)alkyl-hexaazaperhydropyrenes **84**. The intermolecular heterocyclization of 1,4,5,8-tetraazadecalin **57** with *N,N-bis*(methoxymethyl)-*N*-alkylamines in the presence of SmCl_3_ × 6H_2_O as the catalyst or the NiCl_2_ × 6H_2_O-catalyzed ring transformation reaction of 1,3,5-tricycloalkyl-1,3,5-triazinanes with tetraazadecalin led to target azapolycycles (Figure 20).

A similar pathway starting from a mixture of 2,6(7)-dimethyl-1,4,5,8-tetraazadecalin regioisomers was used [44] for the selective synthesis of 4,9-dimethylperhydro-2,3a,5a,7,8a,10a-hexaazapyrenes **85** (Figure 21). The primary antimicrobial screening showed that 2,7-dicyclopentyl-4,9-dimethylperhydro-2,3a,5a,7,8a,10a-hexaazapyrene has a fungicidal activity against *Cryptococcus neoformans var. Grubii* and an antibacterial activity against *Staphylococcus aureus*.

In continuation of these studies, a selective synthesis of 2,7-disubstituted 4,9-dimethyl-perhydro-2,3a,5a,7,8a,10a-hexaazapyrenes **86** and **87** was implemented (Figure 22) by multicomponent condensation of adamantylamines [44] and (het)arylamines [45] with formaldehyde and isomeric tetraazadecalins in the presence of Ymmm zeolite in the H-form or ytterbium (III) chloride.

The use of available secondary metabolites with pronounced pharmacological activity as the starting compounds for the subsequent chemical transformations is an efficient approach to the synthesis of semisynthetic drugs. An example is the synthesis of dimeric terpenoid with a perhydropyrene spacer **88** (Figure 23) [46]. The reaction occurred only in the “strong” acid sites of the H-Ymmm zeolite. High efficiency of the zeolite catalyst is due to the presence of a hierarchical porosity (micro-, meso, and macropores), which provided for the improved diffusion of reactants towards the active sites inside the zeolite pores and the release of the reaction products from the pores to the reaction medium, thus creating conditions for the formation of bulky molecules. Compound **88** exhibited moderate cytotoxic effects against normal cells [human embryonic kidney cell line (HEK293)] and tumor cells [colon adenocarcinoma (HTC-116), human monocytic leukemia (THP-1), breast carcinoma (MCF-7), lung adenocarcinoma (A549), T-cell leukemia (Jurkat), and human neuroblastoma SH-SY5Y)].

## 3. Pentacyclic Saturated Nitrogen-Containing Compounds

### 3.1. Pentacyclanes with One Endocyclic Nitrogen Atom

The shrubs and trees of the *Daphniphyllum genus*, occurring mainly in the subtropical and tropical Asia, are highly valuable sources of natural alkaloids. Several hundred compounds have been isolated from these plants [47,48], but only some pentacyclic compounds, which are shown in Figure 3, had a fully saturated polycyclic core. The key approaches to the preparation of synthetic analogues of *Daphniphyllum* alkaloids have been reported [49].

Out of the *Daphniphyllum* alkaloid family, special mention should be made of the group of compounds with a non-classical structure [50,51]. This group includes, in particular, longeracemine **95** and angustimine **96** (Figure 4). A specific feature of longeracemine **95** is the presence of the C-7/C-9 bond and the 5/6/6/6/5 type ring system. In turn, angustimine **96** is an intramolecular salt containing a non-trivial hexacyclic moiety resulting from the cleavage of the C-6/C-7 bond to the C-6-N bond. Some *Daphniphyllum* alkaloids exhibited anticancer, antioxidant, and anti-HIV activities and stimulated wheat growth. The biological activity of natural *Daphniphyllum* alkaloids has not been exhaustively studied, since pharmacological assays prove to be costly due to low contents of the alkaloids.

The synthesis of azapentacyclic compound **98** (Figure 24) can be successfully performed by the Leuckart–Wallach reductive amination of alicyclic 1,5,9-triketone **97** [52] on treatment with formamide.

### 3.2. Pentacyclanes with Three Endocyclic Nitrogen Atoms

The stereoselective synthesis of the decahydropyrrolo[2,1,5-*cd*]indolizine system has been reported [53]. The 1,3-dipolar cycloaddition of glycine methyl ester and cinnamaldehyde to *N*-phenylmaleimide resulted in the formation of pyrrolidine derivative. The subsequent reaction of the product with cinnamaldehyde and *N*-phenylmaleimide under microwave irradiation (700 W) ended in the formation of dimeric adduct **99**, which cyclized in the presence of iodine monochloride to give decahydropyrrolo[2,1,5-*cd*]indolizine **100** in 38% yield (Figure 25).

A group of researchers reported [54] the synthesis of pentacyclic adducts **101 a–d**. The method included [4+2]-cycloaddition of *N*-substituted maleimides to *N*-benzyl (–)-cytisine and the subsequent hydrogenation on the palladium catalyst. The complete reduction of compound **101 d** enabled the synthesis of saturated triamine **102** in 66% yield (Figure 26). It was shown that (–)-cytisine derivatives of this type exhibited a moderate antiviral activity against the H1N1 serotype influenza A virus [55].

### 3.3. Pentacyclanes with Six and More Endocyclic Nitrogen Atoms

An example of the synthesis of cage pentacyclanes **104** is reported in a study headed by A. Kravchenko [56] (Figure 27). Having replaced glyoxal, which is traditionally used for glycoluril synthesis, by cyclohexane-1,2-dione, the authors obtained derivative **103** with the goal to use it as a synthon for the synthesis of fused polyheterocyclic glycolurils.

A four-step synthesis of methylene-bridged glycoluril dimer **105** in 92% yield has been proposed [57] (Figure 28). The sequence of formation of the octaazapentacyclic compound included diacetylglycoluril alkylation with benzyl bromide in the presence of potassium carbonate, alkaline hydrolysis, and condensation with paraformaldehyde in an acid medium. The final step was the removal of the benzyl groups by treatment with sodium metal in liquid ammonia.

The reaction of disubstituted tetraazaperhydropyrene **70** with ethylenediamine and ammonia resulted in the synthesis of original penta- **106** and polycyclic **107** heterocycles in insignificant yields [32] (Figure 29).

## 4. Hexacyclic Saturated Nitrogen-Containing Compounds

### 4.1. Hexacyclanes with One Endocyclic Nitrogen Atom

A large group of penta- and hexacyclic saturated nitrogen-containing compounds belonging to the family of diterpene alkaloids were isolated from various species of *Aconitum, Delphinium*, and *Garrya* plants (Figure 5). In terms of the chemical structure, they have been classified as C_18_-, C_19_-, or C_20_-diterpene alkaloids without multiple bonds in the molecular core. Most of C_19_-diterpene alkaloids belong to the aconitine and lycoctonine types. The main representatives of C_20_-diterpene alkaloids correspond to atisine and napelline types [58,59,60,61,62,63]. Diterpene alkaloids exhibited a pronounced cytotoxic effect against human tumor cells (A-549, DU-145, MDA-MB-231, MCF-7, HER2, and KB) [64,65]. The anti-inflammatory activity was studied for some representatives of *Aconitum* diterpene alkaloids [66].

In 2014, Japanese researchers headed by T. Fukuyama performed the first total synthesis of (−)-lepenine **113** [67]. The denudatine molecular skeleton of this product represented a hexacyclic system comprising tetradecahydrophenanthrene, a polycyclic system containing a nitrogen atom, and a bicyclo [2.2.2] cage. The multistep synthesis started with the conversion of methyl L-lactate **108** to alcohol **109**, which was transformed to tetracyclic lactone **110** via a sequence of reactions. The subsequent transformations were directed towards the formation of polycyclic system **112** containing a nitrogen atom. The final goal, which was successfully solved, was related to the construction of bicyclo [2.2.2] cage and its subsequent fuctionalization, resulting in the target polycyclane **113** (Figure 30).

### 4.2. Hexacyclanes with Two and Three Endocyclic Nitrogen Atoms

D. Trauner and B.M. Williams proposed [68] a synthetic route to alkaloid (+)-lycopalhine A **121**, which features a complex hexacyclic ring system consisting of one C-6 and two C-5 carbocycles and single pyrrolidine, piperidine, and perhydropyrimidine moieties. L-Glutamic acid ester **114** was used as the starting compound in the multistep synthesis. Using a number of successive transformations, compound **114** was converted to amino ketone **117**, which reacted with aldehyde **118**. The treatment of the resulting imine **119** with L-proline promoted an intramolecular 5-endo-trig Mannich cyclization, resulting in the synthesis of product **120** with the *S*-configuration at C-7. The subsequent intramolecular cyclization enabled the formation of diazapolycyclane **121** (Figure 31). 

A study of Li’s research group [69] describing new representatives of *Myrioneuron* alkaloids with a non-trivial hexacyclic core deserves attention (Figure 6). The structures of the products were determined by a combination of spectroscopic data and a single crystal X-ray diffraction analysis. Compound **123** is the first *Myrioneuron* alkaloid featuring a unique *trans*-decahydroquinoline motif and containing a rare cyano group. Compounds **122 a**,**b** inhibited the hepatitis C virus in vitro.

### 4.3. Hexacyclanes with Six Endocyclic Nitrogen Atoms

The cyclocondensation of 4-aminopiperidine, paraformaldehyde, and tetraazaperhydrotetracene **124** in the presence of the highly acidic Lewatit cation exchange resin [70] enabled the synthesis of *N,N′*-disubstituted hexaazaperhydrodibenzotetracene **125**, the stereoconfiguration of which was not indicated (Figure 32).

A number of studies, addressed the possibility of preparing hexaazaperhydrodibenzotetracenes from tetracyclanes **124 a**,**b** as the starting building blocks, which were obtained from (±)-*cis*-1,2-diaminocyclohexane or (±)-*trans*-1,2-diaminocyclohexane. Heterocyclization [71] of 1,6,7,12-tetraazaperhydrotetracene **124 a**, generated in situ from (±)-*cis*-cyclohexane-1,2-diamine, with *N,N*-di(methoxymethyl)alkylamines or 1,3,5-tricycloalkyl-1,3,5-triazinanes under optimized reaction conditions led to the selective formation of the corresponding 2,9-di(cyclo)alkyl-substituted hexaazaperhydrodibenzotetracenes **126** in 72–82% yields (Figure 33). The hexaazaperhydrodibenzotetracene skeleton has the *trans,trans,anti,trans,trans*-configuration, in which two piperazine and two triazinane rings are in a chair conformation and are fused in the *trans*-form. A distinctive feature of the synthesis based on (±)-*cis*-1,2-diaminocyclohexane is the formation of perhydrotetracenes with *S*,R*,R*,S** relative configuration of the chiral centers at the C-3b, C-7a, C-10b, and C-14a carbon atoms.

The possibility of incorporating amino derivative of the methyl maleopimarate into the hexaazaperhydrodibenzotetracene core (Figure 34) has been reported [46]. Compound **127** exhibits moderate cytotoxic effects against HTC-116, THP-1, MCF-7, Jurkat, and SH-SY5Y tumor cells. The suggested synthesis opens a way to new potentially biologically active hybrid molecules. Combining several moieties with proven pharmacological activity in one molecule is a promising approach to the development of organic compounds with a targeted biological action. This strategy may give rise to compounds possessing a new spectrum of pharmacological properties compared to initial molecules [72].

An efficient one-pot method for the synthesis of previously undescribed 2,9-disubstituted hexaazaperhydrodibenzotetracenes **128** has been developed [73]. The synthesis was based on tetraazaperhydrotetracene **124 b** derived from (±)-*trans*-1,2-diaminocyclohexane (Figure 35). The six-membered carbo and aza rings in the hexaazaperhydrodibenzotetracene moiety exist in the chair conformation. The piperazine rings are *cis*-fused, while the piperazine and cyclohexane rings are *trans*-fused. A structural pecularity of the synthesized hexaazaperhydrodibenzotetracenes is the presence of the *R*,R*,R*,R**-relative configuration of chiral centers at the C-3b, C-7a, C-10b, C-14a carbon atoms and a *cis*-junction of the rings along the C14c–C14d bond. No data on configurations of compounds formed in the reactions of 1,2-diaminocyclohexane isomers with glyoxal are available from the literature. Still, it was reported that the reactions of *trans*-diaminocyclohexane [74,75] with 2,3-dihydro-5,6-dimethylpyrazine give tetraazapolycyclane compounds with *cis*-junction of piperazine rings. The products obtained in the reactions of triethylenetetramine [76] with glyoxalare characterized by similar stereochemistry. It is noteworthy that perhydropyrene polyazapolycycles also have both *trans*- [71] and *cis*-junctions [77] of the piperazine rings.

The broad spectrum of biological activity of the heterosystems with an adamantane moiety [78] attracts the attention of researchers and stimulates the development of efficient methods for the synthesis of new cage molecules. In this regard, the same authors [73] performed the one-pot cyclocondensation of adamantylamines with formaldehyde and tetraazaperhydrotetracene **124 b** in the presence of granulated H-Ymmm zeolite to give previously undescribed diadamantyl-substituted hexaazaperhydrodibenzotetracenes **129** (Figure 36). The hydroxyadamantane-substituted hexaazaperhydrodibenzotetracenew is shown to possess a cytostatic activity (a proliferation-restrictive activity) against six tumor cell cultures (Jurkat, K562, U937, A549, A2780, and T74D). The most pronounced effect was observed in U937 cells. 

The catalytic cyclocondensation of tetraazaperhydrotetracene **124 b** [79,80] with formaldehyde and some (het)arylamines led to the corresponding 2,9-bis(hetaryl) derivatives **130 a,b** (Figure 37). The condensation of heterocycle **124 b** with formaldehyde in the presence of a catalyst involves the stage of formation of tetrakis(hydroxymethyl) derivative. Probably, YbCl_3_ × 6H_2_O or Sm(NO_3_)_3_ × 6H_2_O is coordinated, as a hard Lewis acid, to the oxygen atom of intermediate to give a carbocation. The subsequent nucleophilic addition of the primary (het)arylamine to the carbocation results in di(het)aryl-substituted hexaazaperhydrodibenzotetracenes.

The ability of *N,N,N′,N′*-tetramethylmethanediamine to form new C-N bonds in the catalytic synthesis of heterocyclic compounds has been reported [81]. With this in mind, at the next stage, authors [82] made an attempt to use tetramethylmethanediamine as a synthetic equivalent of formaldehyde in the synthesis of hexaazaperhydrodibenzotetracenes **130 a,b** (Figure 38). The condensation of tetraazaperhydrotetracene **124 b** with *N,N,N′,N′*-tetramethylmethanediamine and substituted anilines was performed in the presence of NiCl_2_ × 6H_2_O as the most active catalyst. According to the hard and soft acids and bases principle [83], Ni(II) chloride hexahydrate, being an “intermediate” Lewis acid, is preferably coordinated to the intermediate *N*-donor ligand to give the most stable acid-base complex.

Hexaazaperhydrodibenzotetracenes **130 a,b** were evaluated for cytotoxicity in a panel of human cancer cell lines (HepG2, HTC116, SH-SY5Y, MCF-7, A549, Jurkat, THP-1) and in tentatively normal cells (HEK293) by the MTT assay [82]. Tetracenes containing 3-methoxyphenyl and 4-nitrophenyl substituents exhibited a moderate cytotoxicity against tentatively normal HEK293 cells only. The compound containing a pyridine substituent was most active amongst the series of substances studied, inhibiting the viability of cancer and non-cancer cells with IC_50_ values ranging from 25 to 52 *μ*M, depending on the cell line tested.

## 5. Heptacyclic Saturated Nitrogen-Containing Compounds

Only a few examples of the synthesis of heptacyclanes with endocyclic nitrogen atoms can be found in the literature. Thus the synthesis of methylene-bridged glycoluril dimer **132** has been proposed [84]. Glycoluril dimer was prepared in two steps. First, a precursor of glycoluril dimer was synthesized and then two precursor molecules **131** were linked via the condensation with formaldehyde (Figure 39).

## 6. Conclusions

The scope of applicability of nitrogen compounds is ever expanding. The molecular cores of *N*-polycyclic compounds possess high structural diversity and attract interest in relation to the search for new medicinal agents. In this review, we considered and described the modern trends in the chemistry of saturated azapolycyclic molecules with promising biological properties such as anticancer, anti-inflammatory, antibacterial, antifungal, antiviral, and other properties. On the one hand, the development of synthetic routes to new original polycyclic systems and the targeted modification of polyheterocycles substantially expand the potential for the synthesis of these compounds. On the other hand, it is of considerable interest to incorporate pharmacologically significant groups and natural metabolites with pronounced biological activity into the polycyclic core in order to obtain new classes of hybrid molecules as promising precursors for the development of modern medications for the treatment of socially significant diseases. Analysis of state-of-the-art studies addressing the synthesis of various saturated *N*-polycyclic molecules confirms their huge potential and implies a broad scope of practical application of these products.

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
