# Peer review of "Recent Advances in the Chemistry of Saturated Annulated Nitrogen-Containing Polycyclic Compounds"

_ijms, 2022, doi:10.3390/ijms232415484_

Round 1

Reviewer 1 Report

- Line 27: adjust space between words.

- The authors need to add application, if any, of all saturated annulated nitrogen-containing polycyclic compounds listed in the review and not some of them in order to emphasize their importance.

- Reference to quote: https://doi.org/10.1002/ejoc.202200432

Author Response

Dear Editor,

The Authors express their sincere gratitude to the Reviewers for their work on evaluating the present article and for their suggestions. We send you a revised version of the manuscript “Recent advances in the chemistry of saturated annulated nitrogen-containing polycyclic compounds” by Kirsanov Victor, Rakhimova Elena and a cover letter detailing the changes we have made in accordance with the Reviewers' comments. 

Reviewer 2 Report

Kirsanov and Rakhimova proposed a review article entitled “Recent advances in the chemistry of saturated annulated nitrogen-containing polycyclic compounds” furnishing a complete overview on the synthesis and related properties of saturated annulate nitrogen-containing polycyclic compounds published over the last 20 years. The scientific soundness of the manuscript is overall excellent as well as its organization, allowing an ease and full approach to a huge chapter of organic synthesis. At the end of the manuscript, the reader gets the impression of dealing with an almost encyclopedic work. The number of citations is appropriate as well as the number of self-citations. As such, I congratulate the authors for this nice work and recommend this manuscript for a publication in IJMS upon an overall brief and general proof-reading. The only think I feel to suggest is, when a synthetic protocol/approach is detailed into the main text, to write the name of the group or the researchers involved instead of writing “French researchers discovered…Italian groups found…” and so on. In this way, the reader gets quicker who is doing what.

Author Response

(The authors gave the same response as above.)
